# Puppet Dynasty Recognition System Based on MobileNetV2

**DOI:** 10.3390/e26080645

**Published:** 2024-07-29

**Authors:** Xiaona Xie, Zeqian Liu, Yuanshuai Wang, Haoyue Fu, Mengqi Liu, Yingqin Zhang, Jinbo Xu

**Affiliations:** 1Art College, Northeastern University, No. 11 Lane, Wenhua Road, Heping District, Shenyang 110819, China; xiezitong@126.com; 2College of Sciences, Northeastern University, No. 11 Lane, Wenhua Road, Heping District, Shenyang 110819, China; lmengqi1104@163.com (M.L.); 13398721708@163.com (Y.Z.); 3Institute of Artificial Intelligence, Beihang University, No. 37 Xueyuan Road, Haidian District, Beijing 100191, China; wangys_neu@126.com; 4School of Mechanical Engineering and Automation, Northeastern University, No. 11 Lane, Wenhua Road, Heping District, Shenyang 110819, China; 20234629@stu.neu.edu.cn

**Keywords:** object detection, deep learning, dynasty identification, convolutional neural network

## Abstract

Traditional image classification usually relies on manual feature extraction; however, with the rapid development of artificial intelligence and intelligent vision technology, deep learning models such as CNNs can automatically extract key features from input images to achieve efficient classification. This study focuses on the application of lightweight separable convolutional neural networks in domain-specific image classification tasks. In this paper, we discuss how to use the SSDLite object detection algorithm combined with the MobileNetV2 lightweight convolutional architecture for puppet dynasty recognition from images—a novel and challenging task. By constructing a system that combines object detection and image classification, we aimed to solve the problem of automatic puppet dynasty recognition to reduce manual intervention and improve recognition efficiency and accuracy. We hope that this will have significant implications in the fields of cultural protection and art history research.

## 1. Introduction

Cultural relic dynasty identification using the SSDLite object detection algorithm is a cutting-edge research topic that integrates computer vision, cultural relic protection, and cultural heritage management. Cultural relics, as historical witnesses, carry rich cultural information and value, so their accurate dynasty identification is of great significance for historical research, cultural relic protection, and exhibition evaluation. However, traditional cultural relic dynasty identification usually relies on manual analysis and professional knowledge, which is relatively subjective, limited, time-consuming, and sometimes inefficient. The development of computer vision and artificial intelligence technology has opened the possibility of applying automated and data-driven methods in cultural relic dynasty identification.

In recent years, convolutional neural networks in computer vision tasks have developed rapidly, driven by research into image classification and object detection. For example, algorithms applied in this field include SSD [1], R-CNN [2], FastR-CNN [3], yolov3 [4], and DSSD [5], and practical applications include medical treatment [6], clothing [7], waste classification [8], content filtering [9], food [10], agriculture [11], language [12], industry [13], autonomous driving [14], and road detection [15]. However, the huge storage and computing requirements hinder the application of CNNs in embedded systems with limited power or resources. For this reason, the lightweight neural network MobileNetV2 [16] has attracted increasing attention. For instance, lightweight neural networks have been applied to agricultural pest detection [6], and good results have been achieved.

In recent years, MobileNetV2 has gained a lot of attention in face recognition, and many corresponding studies have been published. Based on MobileNetV2, Chen S. et al. [17] proposed a particularly efficient CNN model, mobilefacenet, which uses less than 1 million parameters and is specifically customized for high-precision real-time face verification on mobile and embedded devices. The fastest mobilefacenet has a reference time of 18 milliseconds on a mobile phone. Compared with mobile CNNs, which used to be the state-of-the-art technology, the efficiency of MobileFaceNets has significantly improved face verification. Based on the improvement in MobileNetV2, Y. Zhang et al. [18] developed a superior masked face recognition system. They substituted average pooling in the attention module with depthwise separable convolutions and replaced ReLU activation functions with PReLU, enhancing normalization operations. The authors further introduced an improved dual attention module to redistribute weight parameters effectively in bottleneck areas. U.karni et al. [19] proposed the use of the deep learning architecture MobileNetV2 in solving the key problem of face recognition. Using the CNN and MobileNetv2 architecture, 15 key points of a given face image can be detected. Q.Wang et al. [20] constructed a lightweight model, AMSSD, based on SSD reconstruction, in which the MobileNetV2 lightweight network was used in place of the SSD base network in an attempt to reduce the number of parameters. M.U. Khan et al. [21] used the MobilenetV2-based transfer learning method to study how to collect knowledge in a specific dataset and transfer information from the ImageNet dataset (14,197,122 images) to the Kaggle sentiment dataset (36,082 images) to improve face recognition performance.

Against this backdrop, this study aimed to explore the potential applications of lightweight separable convolution technology in puppet dynasty recognition, focusing on SSDLite, a deep learning model that combines object detection and classification tasks. Given its excellent computational efficiency and accuracy, the SSDLite framework has significant advantages in dealing with complex visual problems. Thus, we chose to apply it to automatically recognize puppet figures in different historical periods. The purpose of this study was to develop an automatic classification system for cultural relics that could effectively extract and analyze the facial features of figurines and then accurately distinguish the styles of different eras. With this research, we hope to provide novel technical support for cultural heritage conservation, art history research, and relevant disciplines that will have significant methodological implications.

This study is of practical significance. Firstly, the employment of the SSDLite model in cultural relic dynasty identification not only improves identification accuracy and efficiency but also eases the burden on experts by reducing subjective judgments and errors. Secondly, deep learning technology makes it possible to excavate and analyze hidden information in a large amount of cultural relic image data, providing technical support for cultural relic identification, cultural heritage management, and archaeological research. In addition, this study enriches this field of interdisciplinary study and advances both the theory and practice of cultural heritage conservation. Therefore, the main outcomes of this study will help develop the entire cultural heritage conservation cause.

In this paper, we hope to provide new insights and technical support for automatic cultural relic dynasty identification, inject new vitality into cultural relic protection and historical research, and contribute to the inheritance and promotion of human history and civilization.

This paper is structured as follows: After discussing SSD and MobileNetV2, we also outlines the SSDLite model in Section 2. The experimental results are presented in Section 3. We discuss the experiment in Section 4 and the conclusions are drawn in Section 5.

The following points summarize the contributions of this paper:

Firstly, this paper presents a lightweight separable convolution technique for the dynasty recognition of puppet pictures. In this framework, the advanced deep learning model was introduced, and the puppet dynasty recognition features were adjusted and optimized to realize the efficient classification and recognition of puppet faces.

Secondly, this study integrates target detection and classification methods. In this study, object detection technology, i.e., the SSDLite model, was used to locate and extract the face region in puppet images. Then, the extracted facial features were classified and recognized by the MobilenetV2 model to effectively identify the puppet dynasty.

Thirdly, in this study, deep learning technology was successfully applied to the automatic classification and recognition of puppet dynasties, reducing the need for expert experience and manual feature extraction. More importantly, this study improves the accuracy and reliability of classification and recognition.

## 2. Materials and Methods

### 2.1. SSD Algorithm

The SSD algorithm is a fast and accurate object detection algorithm, which uses a multi-scale feature graph for detection. Using a small convolution check on the feature graph to predict the category score and box offset of the anchor boxes of a series of fixed sets, the prediction of different scales on the feature graph and different aspect ratio boxes is achieved. In addition, the SSD network can be trained end-to-end and achieve high accuracy, i.e., the use of low-resolution input pictures maximizes efficiency while minimizing the accuracy cost.

The SSD algorithm is a kind of one-stage target detector. Based on the VGG16 model, the SSD network adds a convolutional layer on top of the model to obtain more feature maps for detection. The SSD algorithm encourages the combination of feature maps of different scales with anchor boxes of different sizes. Through the category prediction of each anchor box and boundary box regression, it identifies different sizes and proportions of the target. More specifically, the SSD algorithm introduces multiple convolutional feature layers in the last few layers of the convolutional neural network, and each layer is associated with the anchor box and outputs predictive information.

Each cell on the feature gap has a small size, so it is particularly applicable to smaller objects, while those on the receptive field have a large size and are suitable for larger objects. Therefore, the SSD network structure can extract feature information of different scales so as to realize target detection of different sizes. Generally speaking, target detection includes three phases. During the prediction phase, the SSD network architecture adopts the concept of an anchor box. The anchor boxes are learned from the dataset during the training phase, and each anchor box corresponds to the location and size of the target. During the prediction phase, the network classifies and regresses each anchor box to determine the location and size of the target. Through the joint optimization of the training phase and the prediction stage, the SSD network structure can improve the accuracy and speed of target detection.

Figure 1 shows the SSD network architecture. The SSD pipeline begins with a series of convolutional operations that successively extract features from the input image. Departing from traditional methods, in the region where object detection is required, the SSD directly generates its output through additional convolutions. The number of channels in these convolutional layers is determined by the combined total of anchor box quantities and the number of object classes. Specifically, this quantity is given by (the number of anchors × (the number of classes + 4)). After feature extraction, the SSD object detection algorithm generates 8732 prior boxes, which are then subjected to non-maximum suppression (NMS) to identify the optimal ones.

### 2.2. MobileNetV2 Network

As a lightweight convolutional neural network, MobileNetV2 has a significantly reduced model size compared to its predecessor, MobileNetV1, while maintaining and even improving model accuracy. One core innovation of the network is the use of a reciprocal residual structure and linear bottleneck technology. Another is the introduction of a ReLU6 activation function to improve the nonlinear expression ability and save computing resources.

Depthwise convolution is a key method to reduce computational complexity in MobileNetV2. This convolution method divides traditional convolution into two steps: deep convolution, which only performs K×K multiplication between feature graph channels, maintaining the space dimension unchanged, and pointwise convolution, which uses a 1 × 1 convolution kernel to integrate the information of each channel to generate a new feature map. This decomposition greatly reduces the amount of computation and the number of model parameters while maintaining a high classification performance.

In the reciprocal residual structure, the number of channels is first expanded by a 1 × 1 convolution, and then features are extracted by a 3 × 3 depth separable convolution before using a 1 × 1 convolution to compress the number of channels again. The small number of channels at both ends makes the computation cost of channel expansion and compression relatively low, which is opposite to the ascending and descending order of classical residual structures. This design can maintain accuracy while reducing the computational requirements. Furthermore, the application of depthwise convolution further reduces the computational complexity and parameter size.

In addition, MobileNetV2 solves the problem of disappearing gradients in deep neural networks using residual connections, enabling deeper models to be trained. The linear bottleneck module is another advantage of the network architecture, consisting of alternating stacks of depthwise convolution and pointwise convolution. Within each module, pointwise convolution is performed first, then 3 × 3 deep convolution is used, and finally, pointwise convolution is applied again so that deep feature extraction can be achieved without changing the size of the feature map.

The linear bottleneck structure also emphasizes the use of linear activation functions in the bottleneck module. This can retain more original information with lower computational complexity than nonlinear activation functions such as ReLU and sigmoid and avoid information loss caused by nonlinear operations. Specifically, a linear activation function is a specific targeted approach to output while keeping the input constant. It aims to avoid issues such as vanishing gradients. Combined with the reciprocal residual structure, MobileNetV2 ensures excellent model performance while efficiently reducing the amount of computation and the number of parameters.

Figure 2 illustrates the inverse residual of MobileNetV2. In the reciprocal residual, 3 × 3 standard convolution is replaced by deep convolution because the latter can greatly reduce the computational cost. Therefore, appropriately increasing the number of channels can produce better results. In summary, 1 × 1 convolution is first used to increase the number of channels. This is followed by 3 × 3 deep convolution. Finally, 1 × 1 point-by-point convolution is used to reduce the number of channels.

### 2.3. Network Structure Optimization

SSDLite optimizes the underlying network in SSD networks. It uses MobileNetV2 to reduce the complexity of the model. With fewer parameters and computations, these lightweight network structures can ensure high accuracy in target detection tasks. In addition, SSDLite further simplifies the network structure by reducing the number of channels in the feature map and using a small number of prediction layers. SSDLite removes some of the layers from the original network and introduces separable convolution and other lightweight components to further reduce computation and memory requirements.

Figure 3 shows the convolutional layer comparison for SSD and SSDLite. As can be seen, each box represents a convolution layer or a deep convolution layer. SSD first uses 1 × 1 convolution to reduce the number of channels in the convolution layer and then uses 3 × 3 convolution for feature extraction. In SSDLite, the standard convolution is replaced by a depth separable convolution, which splits the traditional convolution operation into two micro-operations: a deep convolution and a pointwise convolution, and greatly reduces the number of calculations and parameters. PW and DW refer to pointwise convolution and deep convolution, respectively.

Taking Figure 3 as an example, the advantages of depthwise separable convolutions over standard convolutions were calculated. The standard convolutional layer has four filters, and each filter package contains three 3 × 3 kernels. Therefore, the number of convolutional layer parameters (1) is 108, and the calculation is 972. In the depth separable convolution, a filter of the deep convolution contains only one 3 × 3 kernel, so the number of convolution layer parameters (2) is 27, and the calculation is 243. Pointwise convolution adopts 1 × 1 convolution. In this step, the number of parameters involved in the convolution layer (3) is 12, and the calculation is 108. The total number of parameters in a depthwise separable convolution is obtained by adding the two parts, resulting in 39. The computational cost of a depthwise separable convolution is also derived from summing the two components, giving 351. According to the computed results, adopting depthwise separable convolutions provides obvious advantages. For example, depthwise separable convolutions significantly decrease the number of parameters and reduce the computational complexity in comparison to standard convolutions.

In addition to these specific computational examples, we further demonstrate the advantages of depthwise separable convolutions by referencing the formulas presented in Kang H.-J.’s paper [22]. As the core component of a CNN, the convolutional layer is based on a two-dimensional convolution. The layer assumes *N* input feature maps g and produces i output features maps ***f*_1_** with kernels of size *K* as follows: (1)$$f_{1}(x,y,i)=\sum_{j=0}^{N-1}\sum_{m=0}^{K-1}\sum_{n=0}^{K-1}w(i,j,m,n)\times g(j,S\times x+m,S\times y+n)+bias(i)$$
where *S* represents the stride; w() denotes the weight functions, and bias() signifies the bias offsets for individual output feature maps.

MobileNet replaces a normal convolutional layer with a combination of depthwise and pointwise convolutional layers. The depthwise convolutional layer calculates the output data by individually applying a *K* × *K* size kernel to each input channel as follows:(2)$$f_{2}(x,y,j)=\sum_{m=0}^{K-1}\sum_{n=0}^{K-1}w(j,m,n)\times g(j,S\times x+m,S\times y+n)+bias(j)$$

According to the provided formula, it is clear that depthwise separable convolutions offer significant advantages over standard convolutions in terms of improved parameter efficiency and reduced computational cost. This architectural innovation not only reduces the memory requirements but also enables faster computations. This makes it particularly suitable for scenarios where resource constraints are a concern without compromising model performance to a large extent, such as in mobile or embedded systems.

In order to implement this model, the following modifications were made to the SSD algorithm:Input Resizing: a more universal input size (320 × 320) was adopted, thereby circumventing the issue of inconsistent padding between TensorFlow and PyTorch when using a 300 × 300 input;C4 Feature Extraction: instead of extracting the C4 feature from within an inverted residual block in the backbone, the refined operation was conducted after the backbone structure itself;Anchor Settings Update: given the change in input resolution, the anchor settings were modified to shift the original utility to accommodate new dimensions;Initialization and Batch Normalization Tweaks: Inspired by the setup in TensorFlow Model Zoo, Truncated Normal initialization was employed. Furthermore, the epsilon (eps) and momentum parameters of the batch normalization layer were adjusted to achieve better performance;Optimizer Configuration: The SGD Momentum optimizer with an initial learning rate of 0.015 was chosen to refine the configuration. Additionally, the weight was set at 4.0 × 10^−5^, and a Cosine Annealing learning rate schedule was utilized.

Through these adjustments, the SSD algorithm was tailored to our specific requirements, enhancing its compatibility, stability, and optimization potential.

### 2.4. Prior Box Optimization

The SSDLite algorithm employs the anchor box methodology, where a set of anchor boxes are predefined as default candidate regions, thereby further reducing computational complexity. SSDLite retains the multi-task loss function from SSD, which encompasses both classification and localization losses. Moreover, akin to SSD, the SSDLite model adopts similar rules for generating default bounding boxes by predefining a collection of prior boxes with varying shapes and sizes at different feature levels, thus ensuring target coverage across a spectrum of dimensions and aspect ratios.

Figure 4 shows a detailed prior box structure adopted by the SSDLite model in the target detection process. The prior box is generated from multi-scale feature maps and optimized by the non-maximum suppression (NMS) algorithm after the prediction.

In SSD and SSDLite, the location and size of an object are predicted using the given prior boxes for each location. SSD extracts the feature maps of six scales for object detection, from 1/8 to 1/256. The prior boxes for the first, fifth, and sixth scale feature maps have the sizes and aspect ratios described under Type 1 in Table 1. For the other scale feature maps, the sizes and aspect ratios are described under Type 2. In contrast, SSDLite uses Type 3 prior boxes for the first scale feature maps and Type 2 prior boxes for the other feature maps.

According to the prior box configurations, more prior boxes are applied in SSD than in SSDLite for the first extracted feature maps. Moreover, SSD’s scale is small, and its resolution is high.

### 2.5. Combined Target Detection and Classification

In traditional computer vision tasks, object detection and classification are often viewed as two separate processes. Object detection aims to determine the location and size of objects in an image, while classification focuses on identifying specific categories of those objects. However, recent studies [23] have shown that the mixed method approach (i.e., the integration of object detection and classification) can significantly improve classification accuracy and efficiency.

Firstly, by conducting target detection first, the background interference can be diminished. The function of the object detection stage is to locate and segment the target objects in the image so that the classification stage can focus more on analyzing the features of these objects rather than being affected by irrelevant background information.

Secondly, the integration of object detection and classification produces better performance when handling multi-object scenarios. In images containing various types of objects, an isolated classification method does not always accurately identify and classify all objects. By contrast, object detection pre-treatment, which segments the image and locates each object, contributes to precise object classification, thus improving classification performance in complex scenes.

Finally, as two separable but compatible modules, object detection and classification can enhance the modularity and scalability of the system. This design makes it possible to optimize and improve each module while maintaining the flexibility and adaptability of the system.

Figure 5 shows the network structure in the SSDLite model applied in the current study. In this model, the input image first goes through the MobileNetV2 feature extraction stage. The extracted features are used for both multi-scale feature map generation and the next layer’s feature extraction. MobileNetV2 was chosen because of its lightweight characteristics. Thereafter, SSDLite enters the auxiliary stage, where multi-scale feature map generation and feature extraction are also performed. Unlike SSD, SSDLite applies a set of deep convolution kernels to predict class probabilities and bounding box position offsets, further reducing computational costs. The final generated multi-scale feature map enters the prediction layer and passes through the NMS algorithm, and the final prediction box result is produced.

## 3. Results

### 3.1. Dataset

The Puppet Dynasty Identification dataset was applied in the current study. To provide an empirical basis for puppet dynasty identification, the dataset was targeted and compiled beforehand. The dataset contains 133 samples: the training set contains 100 samples, and the validation set contains 33 samples. Table 2 summarizes the distribution of samples.

The selected dataset was trained, and evaluation metrics were used to evaluate the experimental results. Thereafter, comparisons were made with other models to demonstrate the feasibility and effectiveness of the selected model.

Table 3 shows the experimental platform used, which is based on the Pytorch 2.0.0 framework. The training environment for the model was PyTorch 2.0.0, Python 3.8 (ubuntu20.04), Cuda 11.8, an NVIDIA A16 graphics card, RTX 4090 GPU, 24 GB graphics memory, and a 16 vCPU AMD EPYC 9654 96 Core Processor on a rented cloud server.

Figure 6 shows several samples from the Puppet Dynasty Identification dataset, including the Republic of China Era (1912–1949) and Qing Dynasty (1644–1911) puppets.

After compiling the dataset, data preprocessing operations were performed. Measures such as standardization, normalization, and data augmentation were adopted.

Specifically, in order to enhance the diversity of the dataset and improve the generalization ability of the model, data augmentation processing was performed. Various image transformation methods, such as rotation, scaling, cropping, and flipping, were used, and image attributes, such as brightness, contrast, and saturation, were adjusted to generate new image samples.

Figure 7 shows the face bounding boxes obtained during object detection.

### 3.2. Evaluation Index

The mean Accuracy Precision (mAP), a widely used evaluation criterion, was employed in the current study to measure performance.

Accuracy represents the ratio of all correctly predicted boxes to all boxes finally output by the network, defined as follows:(3)$$P=\frac{TP}{TP+FP}$$
where *TP* is the number of positive samples correctly identified, and *FP* is the number of negative samples incorrectly identified as positive.

The recall rate represents the ratio of all correctly predicted boxes to all true boxes, defined as follows:(4)$$R=\frac{TP}{TP+FN}$$
where *FN* is the number of positive samples incorrectly identified as negative.

*AP* represents the average accuracy of a category with an IoU threshold between 0.5 and 0.95. It corresponds to the area under the *PR* curve and is an indicator of a single category:(5)$$AP=\int_{0}^{1}P(r)dr$$
where *r* represents the recall rate, and P(r) represents the precision value corresponding to different recall rates.

*mAP* represents the average *AP* of all categories, which is defined as follows:(6)$$mAP=\frac{1}{C}\sum_{i=1}^{C}AP_{i}$$
where *C* represents the number of categories in the dataset, and *AP_i_* represents the *AP* at each confidence level. The higher the *mAP* value, the better the model performance.

In the training process, the loss function is an important index to measure the difference between the model prediction result and the real label.

The loss function is calculated using two components from the weighted average and SSD: the first part is the positioning loss *L_loc_*, and the other is the classification confidence loss rate *L_locf_*, defined as follows:(7)$$L(x,c,l,g)=\frac{1}{n}(L_{conf}(x,c)+\alpha L_{loc}(x,l,g))$$

The locating loss *L_loc_* is defined as follows:(8)$$L_{loc}(x,l,g)=\sum_{i\in Pos}^{N}\sum_{m\in \{cx,cy,w,h\}}x_{ij}^{p}\times smooth_{L1}(l_{i}^{m}-{\hat{g}}_{j}^{m})\;smooth_{L1}(x)=\left\{\begin{array}{l} 0.5x\;\;\mathrm{if}|x|<1\\ |x|-0.5\;otherwise \end{array}\right\}$$
where *Pos* is a positive sample set; *N* is the total number of positive sample *Pos* sets; *m*∈{ce,cy,w,h} is the positional parameter of these four values, representing the coordinates and dimensions of the central point; $x_{ij}^{p}$ is the unique identifier; $l_{i}^{m}$ is the predicted value, and ${\hat{g}}_{j}^{m}$ is the true value.

The confidence loss *L_conf_* is defined as follows:(9)$$L_{conf}(x,c)=\sum_{\dot{i}\in Pos}^{N}x_{ij}^{p}log{\hat{c}}_{i}^{P}-\sum_{\dot{i}\in Neg}log{\hat{c}}_{i}^{0}\;while\;{\hat{c}}_{i}^{P}=\frac{\exp(c_{i}^{p})}{\sum_{P}\exp(C_{i}^{P})}$$
where $x_{ij}^{p}$ is the same as the above position loss; ${\hat{c}}_{i}^{0}$ is the classification probability of the background (negative sample); $C_{i}^{P}$ is the classification part, and the fully connected layer of the general network outputs a vector of the *P* class. The sum of all *P* vector values with a length of *P* + 1 is limited to 1 after the vector passes through softmax. The value with the highest confidence is $C_{i}^{P}$. Moreover, $C_{i}^{P}$ is obtained by applying softmax to $C_{i}^{P}$ and represents the probability of class p.

Table 4 shows the performance of the model in terms of accuracy, recall, f1 score, and support. The support column shows the number of images used to train and validate the model. It can be inferred from the table that the model reached the maximum accuracy, recall, and f1 score in each category. The results can probably be attributed to the background in the images not affecting puppet detection. The background of each figure was deleted after target detection, making the dataset suitable for the proposed model.

Figure 8 shows the experimental results of integrated object detection and the image classification function of the SSDLite model. The left subgraph shows the prior box layout used in the target detection phase and its effective positioning of the puppet’s facial features, which significantly enhanced the classification performance of the system. The subdiagram on the right shows the image classification output based on the SSDLite algorithm, and the “ming” category identifies the puppet as being from the Republic of China. This verifies that the proposed SSDLite framework can effectively realize the accurate identification of dynasty features by integrating object detection technology guided by the prior frame via the image classification method.

Figure 9 illustrates the evolving trends of mean Average Precision (mAP) and loss function values during our training process using SSDLite. As observed, the loss function initially exhibits an ascending trend before descending, a phenomenon typically attributed to the model’s learning and adaptation phase in the early stages of training. At this juncture, the model actively adjusts its weight parameters to capture image features effectively, with the increase in loss being inevitable as the model progresses from its randomly initialized state toward the global optimum. As the training iterations deepen, the optimization algorithm gradually converges, leading to a decrease and eventual stabilization of the loss function, signifying improved learning performance. Concurrently, the mean Average Precision (mAP) consistently increases over time, which demonstrates that SSDLite’s learning capability and generalization performance improved in object detection tasks. This upward trend implies that the model became increasingly adept at accurately localizing and categorizing puppet facial dynasty features.

Figure 10 compares the sizes of the SSDLite model and its SSD counterpart. The illustration clearly depicts a significantly smaller footprint (24.18 MB) than the standard SSD model (182.6 MB). This considerable disparity can be mainly attributed to the optimization strategies employed by SSDLite. By incorporating lightweight network architectures and optimization techniques, SSDLite effectively reduces model complexity, resulting in a more compact model size. This optimization not only boosts computational efficiency but also enables SSDLite to exhibit superior performance in resource-constrained environments. Consequently, SSDLite holds significant practical value within the domain of object detection, particularly for applications where resources are limited, such as in mobile devices and embedded systems. Without compromising on performance, the smaller size makes SSDLite an attractive choice for scenarios requiring efficient yet accurate object detection capabilities.

In order to test the model’s generalization ability in practical applications, an additional dataset was selected, and there were varying degrees of differences between the data collection method, feature distribution, and the training set. This was used to effectively measure the adaptability and robustness of the model. Table 5 summarizes the generalization ability test results. As can be seen, an accuracy of 80% was achieved.

Figure 11 shows the model’s classification results. This test was conducted on an untrained dataset. As shown, the model continued to perform well, exhibiting accurate dynasty recognition and classification.

In this experiment, an additional dataset, which had certain differences in terms of data distribution, feature space, and category labels from the training set but was not completely disconnected, was selected as the test set. The purpose of this was to ensure that the test set not only verified the model’s recognition ability for new samples but also reflected changes that might be encountered in real application scenarios.

During this experiment, the principle of separation was strictly followed to ensure that the test data were not directly or indirectly utilized during the model training process. Subsequently, the images from the test set were fed into the trained image classification model one by one, and the model’s prediction results for each image were recorded.

By comparing the actual labels on the test set with the model’s predicted results, the overall performance of the model on this test set was obtained.

In summary, testing on untrained additional datasets not only validates the effectiveness and generalization ability of image classification models but also provides valuable reference information for further model optimization.

Figure 12 shows the performance comparison for SSDLite and several mainstream object detection algorithms, including RetinaNet [24], YOLOX [25], and Faster R-CNN [26]. RetinaNet is notable in that it addresses the issue of class imbalance in object detection through the introduction of Focal Loss and achieves multi-scale object detection by employing a feature pyramid network. This contributes to its outstanding performance. YOLOX, an upgraded version of the YOLO series, optimizes the anchor-free detection mechanism and incorporates advanced training strategies, striking a balance between high accuracy and speed. Faster R-CNN, on the other hand, represents a two-stage detection method that significantly boosts detection capabilities through the application of a Region Proposal Network (RPN), thereby achieving high precision. The experimental results demonstrate that, within the context of puppet dynasty recognition in the current study, the adopted SSDLite model outperformed the other methods. It was modified for fine-grained puppet image classification, making it particularly applicable for this specialized application scenario.

In this comparative experiment, SSDLite’s mAP was 0.63, RetinaNet’s was 0.593; YOLOX’s was 0.614, and Faster R-CNN’s was 0.627. Thus, it can be seen that the model proposed in this study had superior performance.

## 4. Discussion

In this study, lightweight separable convolution technology, specifically, the SSDLite object detection algorithm and MobileNetV2 classification model, was used to conduct an in-depth exploration and empirical research on dynasty recognition using puppet face images. Through a series of experiments and data analysis, the key results obtained in this study are as follows:

Resource efficiency comparison: In terms of model size, MobileNetV2 has a relatively low number of parameters due to the adoption of depthwise convolution technology. It is evident that its object detection component, SSDLite, further optimizes the network structure, greatly reducing the memory and computing resources occupied by the system as compared with other complex models. This demonstrates the advantages of the lightweight design;

Different scale and aspect ratio adaptability: The model in this study exhibited good adaptability when processing face images of puppets of various sizes and orientations. This is due to SSDLite being designed for detection using multi-scale feature maps and the effective extraction of cross-channel information by depthwise convolution in MobileNetV2. Thus, this model demonstrated its ability with diverse and complex facial features characteristic of puppets from different dynasties;

Error analysis and improvement strategy: Although the overall performance of the model was excellent, misidentification of gender in some specific scenarios could not be completely avoided. Through the in-depth analysis of the error samples, it was found that feature expression was insufficient due to the change in lighting conditions, blurred facial details, and special decorative styles. In this regard, it is suggested that the robustness of the model can be further improved by means of data enhancement or the introduction of an attention mechanism in future studies;

Practical utility: Based on the high-precision recognition results obtained in this study, the feasibility of lightweight separable convolution technology in cultural heritage protection, art history research, and related fields has been demonstrated, providing a reliable and efficient computer vision solution for the era identification of puppets and other cultural artworks.

In summary, by combining SSDLite with MobileNetV2, we not only validated the effectiveness of lightweight separable convolution technology in face classification tasks using puppets but also revealed its potential in resource-limited environments, providing a useful technical reference and practical guidance for similar visual recognition tasks in the future.

## 5. Conclusions

Puppet dynasty identification using images is discussed in this study. By combining object detection and image classification methods, we propose a novel and efficient solution to successfully apply SSDLite in this field.

Firstly, we selected MobileNetV2, a lightweight convolutional neural network architecture based on depthwise convolutions. MobileNetV2 is notable for its superior computational efficiency and small model size. Its backward residual structure, linear bottleneck design, and ReLU6 activation function effectively improve model performance and significantly reduce computational complexity and parameter size, making it particularly suitable for resource-limited embedded environments or real-time applications.

Secondly, in order to precisely capture puppet facial features and accurately perform dynasty classification, we integrated the SSDLite algorithm and used MobileNetV2’s powerful feature extraction capability to process the input images. Our model not only locates the puppet facial features but also achieves high-precision category prediction. This end-to-end approach ensures the real-time accuracy of the system and overcomes the computational bottlenecks and memory limitations of traditional methods when confronted with large amounts of data.

The experimental results indicate the significant improvements brought about by integrating lightweight separable convolution technology and the object detection algorithm. As evidenced by the data, this integration represents a breakthrough in puppet dynasty recognition. Moreover, the proposed model demonstrated excellent performance in terms of recognition accuracy, speed, and resource consumption. This study provides a new technical path and practical implications for future vision-based artificial intelligence in the fields of cultural artwork identification and historical research.

A limitation of this study was the assumption of balanced data. Developing techniques to effectively handle imbalanced datasets, which are common in real-world applications, would be a valuable contribution. Our model, while achieving high accuracy, may not be optimized for real-time applications. Future research should focus on developing lightweight and efficient models that can perform classification tasks in near-real-time.

## Figures and Tables

**Figure 1 entropy-26-00645-f001:**
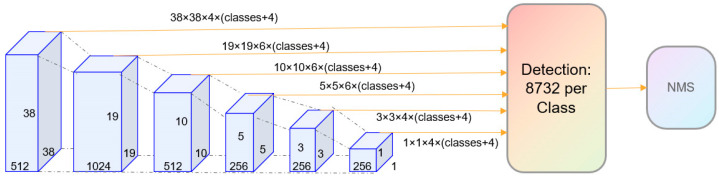
SSD network structure.

**Figure 2 entropy-26-00645-f002:**
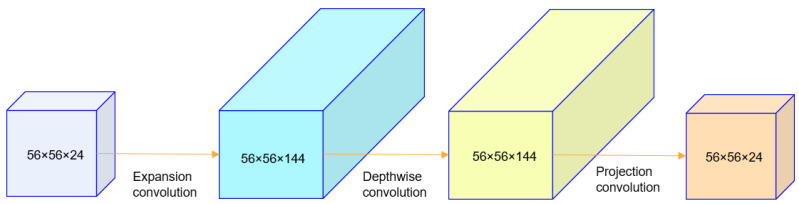
Inverse residual of MobileNetV2.

**Figure 3 entropy-26-00645-f003:**
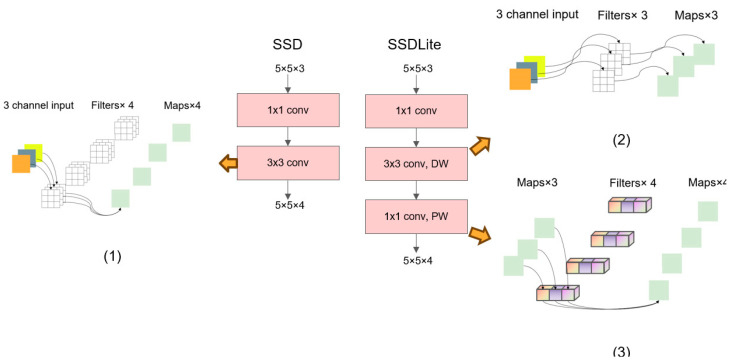
Comparison of SSD and SSDLite convolutional layers, using the 5 × 5 × 3 input as an example. With the same input and output for feature maps, the number of parameters and the computational load for depthwise separable convolutions are approximately one-third of those of standard convolutions. Consequently, under the premise of equal parameter quantities and computational loads, neural network layers employing depthwise separable convolutions can be designed to be significantly deeper.

**Figure 4 entropy-26-00645-f004:**
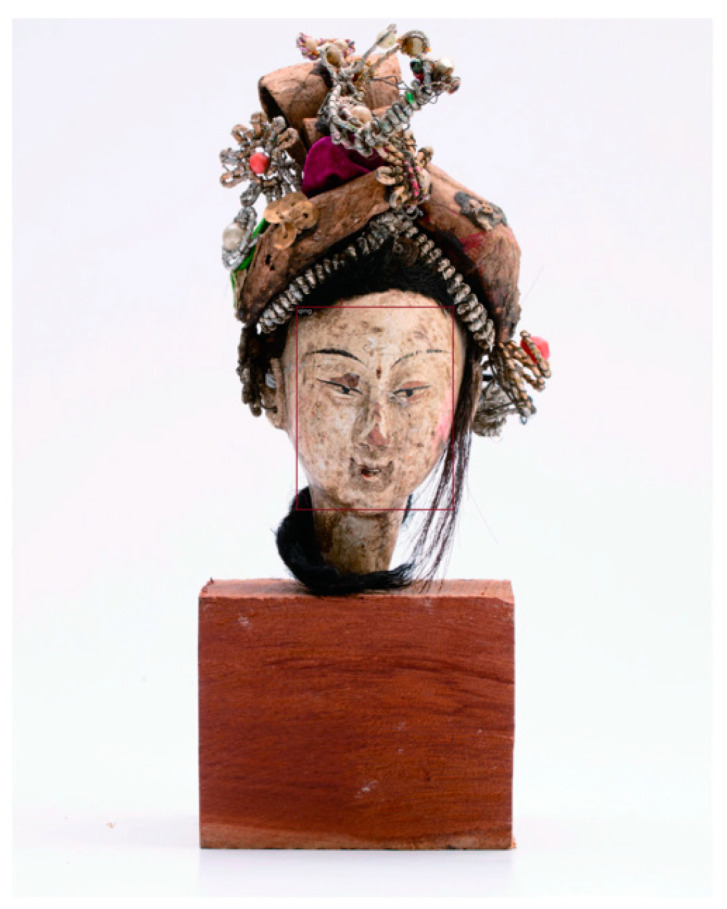
Prior box display.

**Figure 5 entropy-26-00645-f005:**
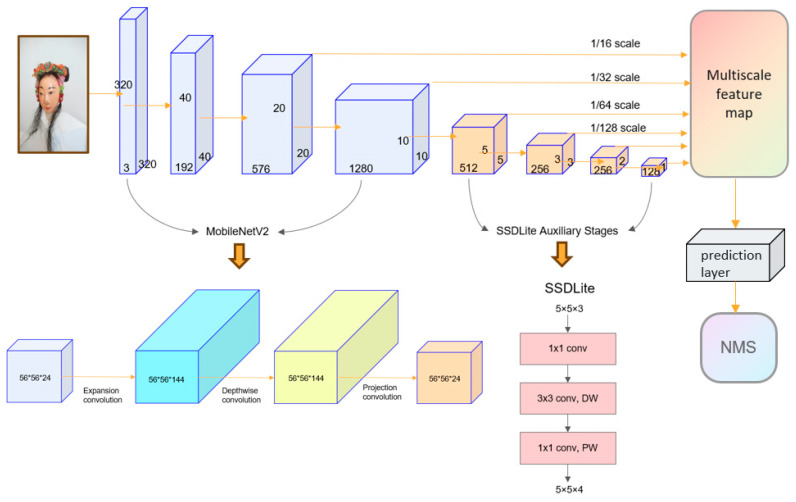
Network structure of SSDLite.

**Figure 6 entropy-26-00645-f006:**
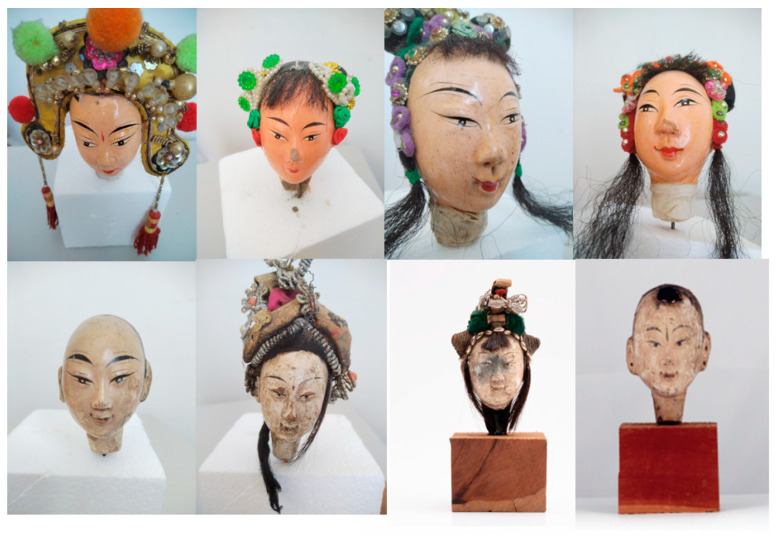
Dataset images. The top row shows the Republic of China Era samples and the bottom row shows Qing dynasty samples.

**Figure 7 entropy-26-00645-f007:**
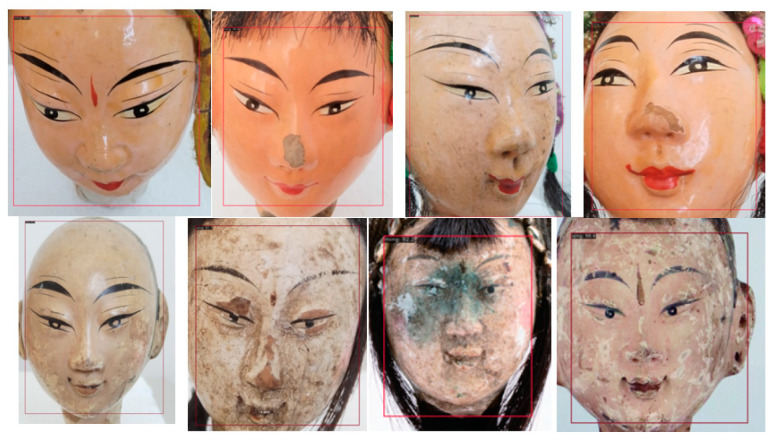
The face bounding boxes.

**Figure 8 entropy-26-00645-f008:**
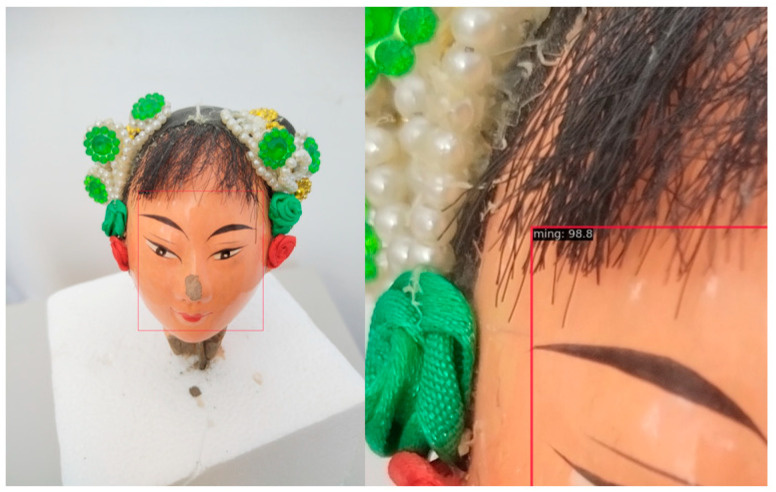
Detection and classification results.

**Figure 9 entropy-26-00645-f009:**
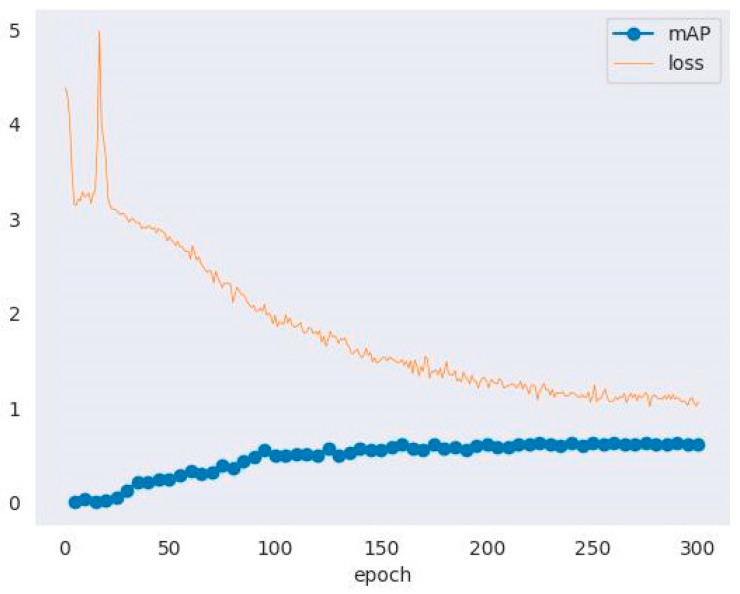
Changes in mAP and loss during training.

**Figure 10 entropy-26-00645-f010:**
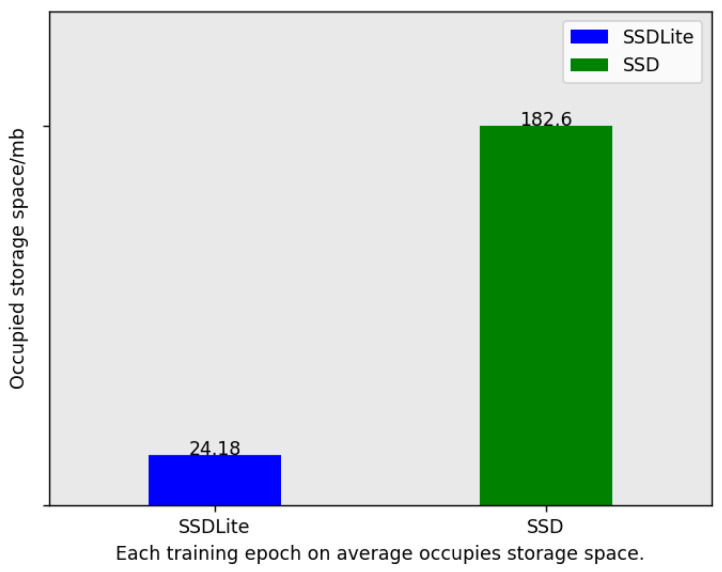
Model size comparison for SSDLite and SSD.

**Figure 11 entropy-26-00645-f011:**
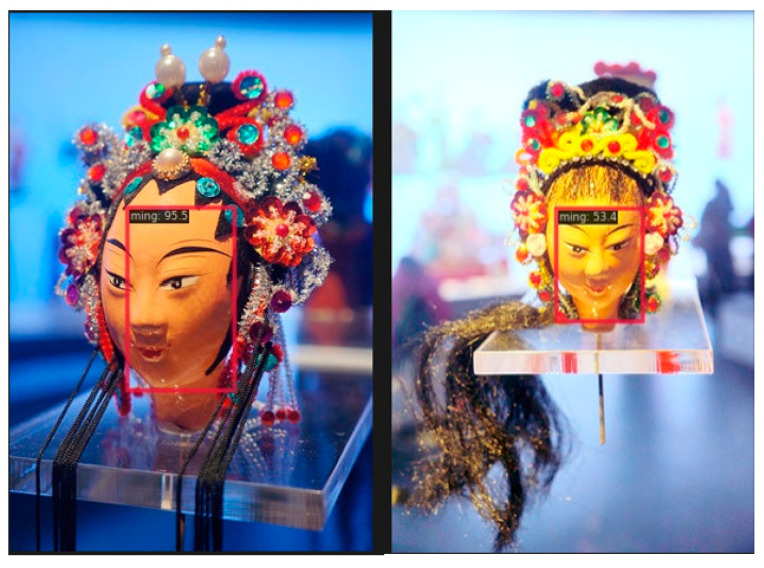
Results using additional datasets.

**Figure 12 entropy-26-00645-f012:**
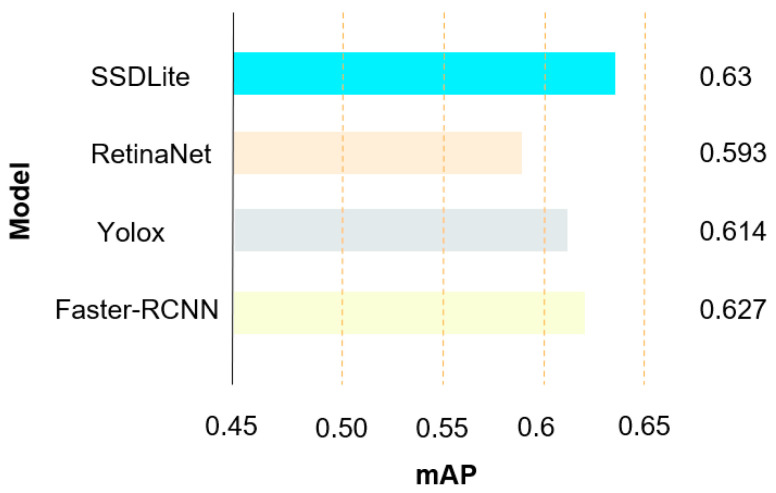
Comparison of mAP for different methods.

**Table 1 entropy-26-00645-t001:** Prior boxes.

Type	(Size, Aspect Ratio)
Type 1	(1,1)$\left(\sqrt{2},1\right)$$(1,2)(1,\frac{1}{2}$)
Type 2	$(1,1)(\sqrt{2}$$,1)(1,2)(1,\frac{1}{2}$$)(1,3)(1,\frac{1}{3}$)
Type 3	$(\frac{1}{2}$$,1)\;(1,2)(1,\frac{1}{2}$)

In this table, size 1 is the reference size, meaning the input image size. Size $\sqrt{2}$ is the geometric mean of the reference size of the current scale and that of the next scale.

**Table 2 entropy-26-00645-t002:** Puppet dynasty identification dataset.

Types	Numbers
Pictures	133
Training samples	100
Test samples	33
Samples of the Republic of China Era	60
Sample number in Qing Dynasty	40

**Table 3 entropy-26-00645-t003:** Experimental platform.

Platform	Version
System	ubuntu20.04
CUDA	11.8
Python	3.8
PyTorch	2.0.0
GPU	RTX 4090
CPU	16 vCPU AMD EPYC 9654 96-Core Processor

**Table 4 entropy-26-00645-t004:** Model validation results.

Type	Precision	Recall	F1 Score	Support
The Republic of China	1	1	1	79
Qing Dynasty	1	1	1	54
Weighted average	1	1	1	133

**Table 5 entropy-26-00645-t005:** Generalization ability test.

Type	Support
The Republic of China	7
Qing Dynasty	3
Correct sample from The Republic of China	6
Correct sample from The Republic of China	2
Accuracy	80%

## Data Availability

The raw data supporting the conclusions of this article will be made available by the authors upon request.

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
