# Peer review of "Puppet Dynasty Recognition System Based on MobileNetV2"

_entropy, 2024, doi:10.3390/e26080645_

Round 1

Reviewer 1 Report

Comments and Suggestions for Authors

The proposed method uses state-of-the-art technologies to address a specific problem in a specialized area, providing benefits in terms of automation, efficiency and reduction of manual intervention. However, it also presents challenges related to implementation, data quality and model robustness. To maximize benefits and minimize limitations, careful attention should be paid to data collection and annotation, as well as continuous optimization of models to ensure optimal performance in real-world scenarios. Moreover:

- Although lightweight architectures like MobileNetV2 are less resource demanding, their effective implementation still requires adequate infrastructure and specialized technical skills.

- The performance of deep learning models strongly depends on the quality and quantity of training data. In the case of puppet images, obtaining a diverse and well-annotated dataset could be a major challenge.

- Puppet images can vary greatly in style, quality and shooting conditions. The model's ability to generalize and provide accurate results across varied and novel images is a crucial consideration.

- Although the method is effective for the specific task of recognizing dynasties on puppet images, its applicability to other areas or types of images might be limited without further modifications and adaptations.

- The use of moment desciptors for the representation and classification of images is very interesting especially for an external comparison

Comments on the Quality of English Language

Extensive editing of English language required

Author Response

Comments1: Although lightweight architectures like MobileNetV2 are less resource demanding, their effective implementation still requires adequate infrastructure and specialized technical skills.

Response 1: Thank you for pointing out this point. We agree with this opinion. Therefore, we have added information about the experimental platform and accurately indicated this change in the revised manuscript, which can be seen on page 9, paragraph 3.

Comments2: The performance of deep learning models strongly depends on the quality and quantity of training data. In the case of puppet images, obtaining a diverse and well-annotated dataset could be a major challenge.

Response 2: Thank you for pointing out this point. We agree with this opinion. Therefore, we have added an introduction to dataset preprocessing before the experiment and accurately pointed out this change in the revised manuscript, which can be seen on page 10, paragraph 1.

Comments3: Puppet images can vary greatly in style, quality and shooting conditions. The model's ability to generalize and provide accurate results across varied and novel images is a crucial consideration.

Response 3: Thank you for pointing out this point. We agree with this opinion. Therefore, we have added an experiment tested on an untrained dataset, which accurately indicates this change in the revised manuscript, as can be seen on page 13, paragraph 2.

Comments4: Although the method is effective for the specific task of recognizing dynasties on puppet images, its applicability to other areas or types of images might be limited without further modifications and adaptations.

Response 4: Thank you for pointing out this point. We agree with this opinion. Therefore, we discussed the performance of the model on different datasets and accurately pointed out this change in the revised manuscript, which can be seen on page 14, paragraph 2.

Comments5: The use of moment desciptors for the representation and classification of images is very interesting especially for an external comparison

Response 5: Thank you for pointing out this point. We agree with this opinion. Therefore, we have supplemented the results of the comparative experiment in the form of text, accurately indicating this change in the revised manuscript, which can be seen on page 15, paragraph

Reviewer 2 Report

Comments and Suggestions for Authors

I want to share some constructive feedback. Overall, I found the content insightful and valuable. However, I noticed a few areas that could benefit from improvement (I attach the PDF with notes and underlined words):

Images: Some of the images require corrections. 

Repetitions: There are instances of repeated words or phrases throughout the article. Consider revising these for clarity.

Missing Words: I noticed a few missing words in certain sections. Please review and fill in the gaps.

Style: While the writing is clear, there are opportunities to enhance the overall style. Consider refining sentence structures and ensuring consistency.

Comments on the Quality of English Language

To enhance the quality of the English language text, it is advisable to make minor adjustments for clarity and precision. Removing any underlining will contribute to a cleaner and more professional presentation. Additionally, the exclusion of the Chinese character will ensure consistency in language use, maintaining the focus on English text quality. These refinements will collectively improve the readability and overall impression of the paper.

Author Response

Images: Some of the images require corrections. 
Response: Thank you for pointing out this point. We have modified the titles of all the images.

Repetitions: There are instances of repeated words or phrases throughout the article. Consider revising these for clarity.
Response:  Thank you for pointing out this point.  We have revised the English expression of the entire article.

Missing Words: I noticed a few missing words in certain sections. Please review and fill in the gaps.
Response:   Thank you for pointing out this point. We have revised the English expression of the entire article.

Style: While the writing is clear, there are opportunities to enhance the overall style. Consider refining sentence structures and ensuring consistency.
Response:  Thank you for pointing out this point.  We have revised the English expression of the entire article.

Round 2

Reviewer 1 Report

Comments and Suggestions for Authors

My comments have been well processed and I therefore suggest acceptance of the paper.

Author Response

Comment: My comments have been well processed and I therefore suggest acceptance of the paper.

Response: We are overjoyed to learn that the reviewer has recommended acceptance of our work with their favorable comments and constructive suggestions. We look forward to your further consideration of the revised manuscript and any additional guidance you may have. Thank you for your continued support and dedication to promoting quality research in our field.
